# Learning to Reconstruct Missing Data from Spatiotemporal Graphs with Sparse Observations

**Ivan Marisca**[*]
[1]The Swiss AI Lab IDSIA,
Università della Svizzera italiana
`ivan.marisca@usi.ch`

**Andrea Cini**[*]
[1]The Swiss AI Lab IDSIA,
Università della Svizzera italiana
`andrea.cini@usi.ch`

**Cesare Alippi**
[1]The Swiss AI Lab IDSIA,
Università della Svizzera italiana
[2]Politecnico di Milano
`cesare.alippi@usi.ch`

## Abstract

Modeling multivariate time series as temporal signals over a (possibly dynamic) graph is an effective representational framework that allows for developing models for time series analysis. In fact, discrete sequences of graphs can be processed by autoregressive graph neural networks to recursively learn representations at each discrete point in time and space. Spatiotemporal graphs are often highly sparse, with time series characterized by multiple, concurrent, and long sequences of missing data, e.g., due to the unreliable underlying sensor network. In this context, autoregressive models can be brittle and exhibit unstable learning dynamics. The objective of this paper is, then, to tackle the problem of learning effective models to reconstruct, i.e., impute, missing data points by conditioning the reconstruction only on the available observations. In particular, we propose a novel class of attention-based architectures that, given a set of highly sparse discrete observations, learn a representation for points in time and space by exploiting a spatiotemporal propagation architecture aligned with the imputation task. Representations are trained end-to-end to reconstruct observations w.r.t. the corresponding sensor and its neighboring nodes. Compared to the state of the art, our model handles sparse data without propagating prediction errors or requiring a bidirectional model to encode forward and backward time dependencies. Empirical results on representative benchmarks show the effectiveness of the proposed method.

## 1 Introduction

Exploiting structure – both temporal and spatial – is arguably the key ingredient for the success of modern deep learning architectures and models. Structure and invariances allow imposing inductive biases to learning systems that act as strong regularizations, thus limiting the space of possible models to the most plausible ones and, consequently, greatly reducing sample complexity. This is the case with spatiotemporal graph neural networks (STGNNs) [1–3] which learn to process multivariate time series while taking into account underlying space and time dependencies by encoding structural spatiotemporal inductive biases in their architectures. In particular, graphs are used to model the presence of spatial relationships which can be thought of as soft functional dependencies existing

---

[*]Equal contribution.

36th Conference on Neural Information Processing Systems (NeurIPS 2022).

among the sensors generating the time series [4]. However, even when spatiotemporal relationships are present, available data are almost always incomplete and irregularly sampled both spatially and temporally. This is definitely true for data coming from real sensor networks (SNs), where the presence of missing data is a phenomenon inherent to any data acquisition or communication process. In this case, a common approach is to reconstruct missing time series observations with simple interpolation strategies before proceeding with the downstream task. However, arguably, this induces a bias in the inference procedure, possibly made even worse by the strong regularization imposed on the model. More advanced methods deal with missing data by autoregressively replacing missing observations with predicted ones, eventually using bidirectional architectures [5, 6] to exploit both forward and backward temporal dependencies. To account also for spatial dependencies, Cini et al. [4] introduced a method, named GRIN, combining a bidirectional autoregressive architecture with message passing graph neural networks [7–10] for spatiotemporal imputation. Despite being the state of the art in imputation, GRIN suffers from the error propagation typical of autoregressive models that bootstrap future inferences from their own predictions. In fact, we argue that the propagation of imputed (biased) values through space and time combined with noisy observations might exacerbate error accumulation in highly sparse data and drive the hidden state of GRIN-like models to possibly drift away. In this paper, we aim at tackling this problem by designing an architecture based on a novel attention mechanism that takes spatiotemporal sparsity into account while learning representations and imputing missing values.

Attention-based models allow distributed representations to emerge by letting each discrete element – representing an entity or point within a structured environment – interact with each other. In fact, these mechanisms for propagating information through structures have been recently linked to collective intelligence [11]. In our context, each token, represented as a node in a sequence of graphs, represents a point in space and time. A trivial way to account for sparsity in this setting is to constrain the attention coefficients of time steps corresponding to missing data to be zero or simply to add an auxiliary token to indicate missing observations; basically, the resulting reconstruction model would behave similarly to a denoising autoencoder, in the same spirit of BERT-like encoders popular in natural language processing [12]. Conversely, we propose a novel architecture exploiting an inter-node sparse spatiotemporal attention mechanism within the neural message-passing framework. In particular, we seek to design an architecture where each processing stage is aligned with the task of reconstructing missing spatiotemporal observations. Our model can be seen as a learned, self-organizing, spatiotemporal propagation process that, as we will motivate throughout the paper, is more apt to the purpose of missing data reconstruction than standard encoder-decoder architectures. In fact, compared with the alternatives discussed so far, our method exploits the aforementioned propagation process to learn a predictive representation for each missing observation by relying only on observed values propagated through the spatiotemporal structure. This approach achieves the twofold objective of avoiding propagating biased representation – typical in the autoregressive framework – and reconstructing observations at arbitrary nodes in the sensor network. In summary, our main contributions are as follows.

1. We introduce a sparse spatiotemporal attention mechanism to learn, from sparse data, representations localized in time and space.

2. We design a novel graph neural network architecture based on the aforementioned spatiotemporal attention mechanism and equipped with inductive biases that make the model tailored for the multivariate time series imputation task.

3. We empirically assess the proposed method, showing how it overcomes the limits of existing approaches, particularly in settings with highly sparse data.

The paper is structured as follows. In Section 2 we discuss related works and contrast them with ours. We formulate the problem of multivariate time series imputation in the context of spatiotemporal graphs in Section 3, and present our approach in Section 4 by providing an in-depth discussion of motivations, design choices, as well as methodological and technical issues. Finally, in Section 5, we report the empirical results on several relevant benchmarks. Conclusions and future works are discussed in Section 6.

## 2 Related works

Multivariate time series imputation is a core task in time series analysis. Besides standard statistical approaches based on linear autoregressive models [13–15] or interpolation [16], methods based on matrix factorization are widely popular [17] and can also incorporate temporal [18] and graph-side information [19]. Deep learning methods are also commonly used in this regard. In particular, deep autoregressive models based on recurrent neural networkss (RNNs) are currently among the most widely adopted methods [20, 5, 6, 21]. In this direction, BRITS [6] is archetypal of several related works which exploit bidirectional RNNs to perform imputation. Notably, differently from a denoising autoencoder, BRITS uses one-step-ahead forecasting as a surrogate task to learn an imputation model while using a simple linear regression layer to incorporate spatial information. Several approaches in the literature, then, rely on generative adversarial neural networks [22] – often paired with RNNs – to generate imputed subsequences by matching the underlying data distribution [23, 24, 21]. Recently, several attention-based imputation techniques have also been proposed [25–27], however, none of these explicitly account for spatial dependencies within the graph processing framework and overlook the spatial dimension of the problem. Other works, instead, address this problem in the context of continuous-time models [28]. Huang et al. [29], in particular, exploit graph representations to model spatial dynamics. The limits of deep autoregressive approaches in data reconstruction have also been tackled by using hierarchical imputation methods [30, 31]. More related to our work, GRIN [4] uses a bidirectional graph RNN, paired with a message passing spatial decoder, to impute time series based on spatiotemporal dependencies. Other graph-based architectures have been used in application-specific settings, such as traffic data [32, 33] and load profiles from smart grids [34]. While GRIN achieves remarkable performance, we argue that spatial regularization might not be enough to prevent error compounding in the states of the recurrent graph network.

The attention mechanism has been exploited in several contexts within the graph deep learning literature, in particular in anisotropic graph convolutional filters [35–38]. Among STGNNs, attention-based architectures have been exploited in time series forecasting [37, 39–41]. In particular, TraverseNet [41] is specially related to our work, since it relies on spatiotemporal autoregressive attention to compute messages exchanged between nodes. One striking example of attention being used successfully to process incomplete data is in pretraining routines for representation learning in natural language processing [12, 42]. Finally, graph neural networks are also popular for reconstructing missing features in static graphs [43–45].

## 3 Preliminaries

We model multivariate spatiotemporal time series as observations coming from a SNs. In a SN every $i$-th node, i.e., *sensor*, acquires a $d$-dimensional $\boldsymbol{x}_t^i \in \mathbb{R}^d$ observation at each $t$-th time step. We denote by $\boldsymbol{X}_t \in \mathbb{R}^{N_t \times d}$ the matrix collecting the measurements of $N_t$ sensors at time step $t$, with $\boldsymbol{X}_{t:t+T}$ being the sequence of $T$ measurements collected in the time interval $[t, t+T)$. We model functional relationships among the sensors as graph edges. Relationships can be often inferred from available side information, e.g., one can extract a graph from the position of each sensor and their reciprocal physical proximity or the structure of the physical system where sensors are placed. In other cases instead, functional dependencies can be inferred directly from data by exploiting some affinity score (e.g., Pearson correlation, Granger causality [46], correntropy [47], etc.), or more advanced techniques (e.g., graph learning methods [48]). We model the extracted relational information with a weighted, possibly asymmetric adjacency matrix $\boldsymbol{A}_t \in \mathbb{R}^{N_t \times N_t}$, in which each nonzero entry $a_t^{i,j}$ indicates the weight of the edge going from the $i$-th node to the $j$-th. While our framework can account for dynamic relationships, we mostly focus on settings where the topology is static, i.e., $\boldsymbol{A}_t = \boldsymbol{A}$ and $N_t = N$. Finally, we assume to have available sensor-level covariates $\boldsymbol{Q}_t \in \mathbb{R}^{N \times d_q}$ that act as spatiotemporal coordinates to localize a point in time and space (e.g., date/time features and geographic location). Note that coordinates $\boldsymbol{q}_t^i$ are assumed available for each node at each time step; in section 4 we discuss how $\boldsymbol{Q}_t$ can be learned as a spatiotemporal positional encoding. We then model observations as a discrete sequence of spatiotemporal graphs, where each graph is a triplet $\mathcal{G}_t = \langle \boldsymbol{X}_t, \boldsymbol{Q}_t, \boldsymbol{A} \rangle$. Note that, in general, we indicate with capital letters network-level attributes, while we use lowercase for the sensor level.

As already mentioned, observations might be incomplete due to faults, partial observability, and costs of the data acquisition process. We model data availability with a binary mask $m_t^i \in \{0, 1\}$ which is

1 if the measurements associated with the $i$-th sensor are valid at time step $t$. Conversely, if $m_t^i = 0$, we consider the measurements $\boldsymbol{x}_t^i$ to be completely missing, with the exogenous variables $\boldsymbol{q}_t^i$ being instead available. Notice that usually, each $m_t^i$ is a realization of random variables with non-trivial distributions exhibiting complex spatiotemporal dynamics and correlations. For instance, a sensor fault may last for more than one time step, or spatially correlated errors may result in missing data in an entire sub-region of the network.

**Multivariate Time Series Imputation**   Given observations $\boldsymbol{X}_{t:t+T}$ with missing values indicated by mask $\boldsymbol{M}_{t:t+T}$, we denote by $\widetilde{\boldsymbol{X}}_{t:t+T}$ the unknown corresponding complete sequence, i.e., the sequence where no observation is missing. Formally, the goal of multivariate time series imputation (MTSI) is to find an estimate $\widehat{\boldsymbol{X}}_{t:t+T}$ minimizing the reconstruction error

$$\mathcal{L}\left(\widehat{\boldsymbol{X}}_{t:t+T}, \widetilde{\boldsymbol{X}}_{t:t+T}, \boldsymbol{M}_{t:t+T}\right) = \sum_{\tau=t}^{t+T} \frac{\sum_{i=1}^{N} \overline{m}_\tau^i \cdot \ell\left(\hat{\boldsymbol{x}}_\tau^i, \tilde{\boldsymbol{x}}_\tau^i\right)}{\sum_{i=1}^{N} \overline{m}_\tau^i}, \tag{1}$$

where $\ell(\,\cdot\,,\,\cdot\,)$ is an element-wise error function and $\overline{m}_\tau^i$ is the logical binary complement of $m_\tau^i$. Notice that, since $\widetilde{\boldsymbol{X}}_{t:t+T}$ is not available, one should find a surrogate objective or simulate the presence of missing data, for which the reconstruction error can be computed.

## 4   Learning representations from sparse spatiotemporal data

The autoregressive approach to reconstruction consists in directly modeling distributions $p\left(\boldsymbol{x}_t^i \,|\, \boldsymbol{X}_{<t}\right)$ and using one-step-ahead forecasting as a surrogate objective to learn how to recover missing observations. To exploit available data subsequent to the target time step, it is common to use a bidirectional architecture, i.e., to mirror time w.r.t. the time step of interest and have a second model for estimating $p\left(\boldsymbol{x}_t^i \,|\, \boldsymbol{X}_{>t}\right)$ [49, 6]. Here we use the shorthand $\boldsymbol{X}_{<t}$ and $\boldsymbol{X}_{>t}$ to denote observations collected before and after time step $t$, respectively. Moreover, a third component $p\left(\boldsymbol{x}_t^i \,|\, \{\boldsymbol{x}_t^{j \neq i}\}\right)$ must be introduced to account for spatial information at each step. Architectures like GRIN, follow exactly this scheme with different components dedicated to model each of these three aspects. While being effective in practice, these approaches can have multiple drawbacks. Besides the computational overhead of having three separate components and the compounding of prediction errors typical of autoregressive models, the modular approach can fall short in capturing global context, as the processing of the structural information is decomposed. Furthermore, integrating the information coming from the different modules is also problematic, yielding further compounding of errors. Finally, in the case of highly sparse observations, the spatial processing should be dealt with special care as propagating information through partially observed spatiotemporal graphs adds another layer of complexity. For instance, simply masking out faulty sensors would compromise the flow of information through message-passing layers [9].

**Model overview**   To address the limitations of existing methods, we act on the problem more directly by aligning the structure of our proposed architecture closely with the reconstruction task. We denote as *observed set* $\mathcal{X}_{t:t+T} = \left\{\langle \boldsymbol{x}_\tau^i, \boldsymbol{q}_\tau^i \rangle \,|\, m_\tau^i = 1, \tau \in [t, t+T]\right\}$ the set of all observations, paired with their spatiotemporal coordinates. Conversely, we name *target set* $\mathcal{Y}_{t:t+T} = \left\{\boldsymbol{q}_\tau^i \,|\, m_\tau^i = 0, \tau \in [t, t+T]\right\}$ the complement set collecting the coordinates of the discrete spatiotemporal points for which we want to reconstruct an observation. We refer to the set of observed and target points of the $i$-th node as $\mathcal{X}_{t:t+T}^i$ and $\mathcal{Y}_{t:t+T}^i$, respectively. Then, for all $\boldsymbol{q}_\tau^i \in \mathcal{Y}_{t:t+T}$, we aim at learning a structured model for

$$p\left(\boldsymbol{x}_\tau^i \,|\, \boldsymbol{q}_\tau^i, \mathcal{X}_{t:t+T}, \boldsymbol{A}\right). \tag{2}$$

Our approach, named *Spatiotemporal Point Inference Network* (SPIN), is a graph attention network for MTSI, designed to learn representations of discrete points associated with nodes of a sequence of spatiotemporal graphs. Given disjoint observed and target sets $\mathcal{X}_{t:t+T}$ and $\mathcal{Y}_{t:t+T}$, SPIN is trained to learn a model

$$f_\theta(\boldsymbol{q}_\tau^i \,|\, \mathcal{X}_{t:t+T}, \boldsymbol{A}) \approx \mathbb{E}\left[p\left(\boldsymbol{x}_\tau^i \,|\, \boldsymbol{q}_\tau^i, \mathcal{X}_{t:t+T}, \boldsymbol{A}\right)\right] \tag{3}$$

for all discrete points $\boldsymbol{q}_\tau^i \in \mathcal{Y}_{t:t+T}$. To this end, SPIN learns a parameterized propagation process where each representation, corresponding to a specific node and time step, is updated by aggregating

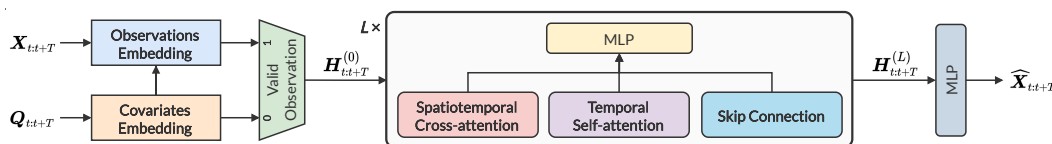

Figure 1: The architecture of SPIN. At first, we encode observations $\boldsymbol{X}_{t:t+T}$ and spatiotemporal coordinates $\boldsymbol{Q}_{t:t+T}$, obtaining initial representations $\boldsymbol{H}_{t:t+T}^{(0)}$. The representations are updated by a stack of $L$ sparse spatiotemporal attention blocks. Final imputations are obtained from $\boldsymbol{H}_{t:t+T}^{(L)}$ with a nonlinear readout.

information from all the available observations acquired at neighboring nodes weighted by input-dependent attention coefficients. Figure 1 shows an overview of the architecture. In the next paragraphs, we will go through each component explaining it in detail and providing motivations for each design choice. We start by describing how we set up the propagation process.

**Sparse spatiotemporal attention**    The core component of SPIN is a novel *sparse spatiotemporal attention* layer used to propagate information at the level of single observations. Indeed, leveraging on the attention mechanism, we learn representations for each $i$-th node at each $\tau$-th time step by simultaneously aggregating information from (1) the observed set of $i$-th node $\mathcal{X}_{t:t+T}^i$ (2) the observed set $\mathcal{X}_{t:t+T}^j$ of its neighbors $j \in \mathcal{N}(i)$. Figure 2 shows a schematic representation of this procedure.

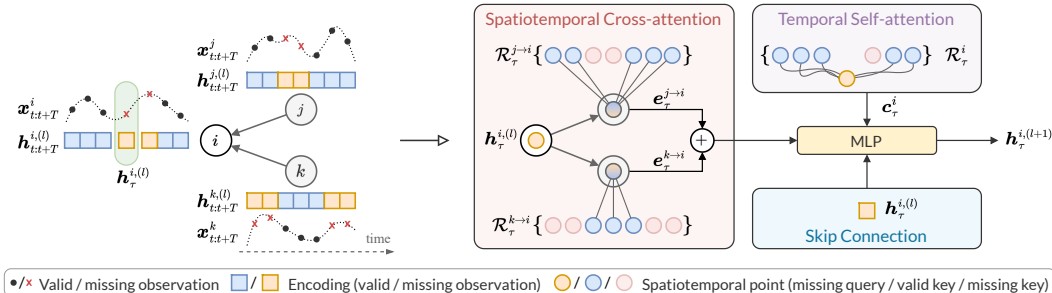

Figure 2: Example of the sparse spatiotemporal attention layer. On the left, the input spatiotemporal graph, with time series associated with every node. On the right, how the layer acts to update target representation $\boldsymbol{h}_\tau^{i,(l)}$ (highlighted by the green box), by simultaneously performing inter-node spatiotemporal cross-attention (red block) and intra-node temporal self-attention (violet block).

Let $\boldsymbol{h}_\tau^{i,(l)} \in \mathbb{R}^{d_h}$ be the learned representation for the $i$-th node and time step $\tau$ at the $l$-th layer. The encoding is initialized as

$$\boldsymbol{h}_\tau^{i,(0)} = \begin{cases} \text{MLP}(\boldsymbol{q}_\tau^i) & \boldsymbol{q}_\tau^i \in \mathcal{Y}_{t:t+T} \\ \text{MLP}(\boldsymbol{x}_\tau^i, \boldsymbol{q}_\tau^i) & \langle \boldsymbol{x}_\tau^i, \boldsymbol{q}_\tau^i \rangle \in \mathcal{X}_{t:t+T} \end{cases} \tag{4}$$

where $\boldsymbol{q}_\tau^i$ indicates a positional encoding, as previously anticipated, and MLP is a generic multi-layer perceptron. The representation is then updated by joint temporal and spatiotemporal attention operations. To describe the inner working of the proposed attention mechanism, we adopt the terminology of Vaswani et al. [50] and indicate as *query* the token for which we want to compute an updated representation, *key* a representation of the source tokens, and *value* the content representation of each token. The next steps involve computations of spatiotemporal messages, i.e., representations computed to propagate information from one discrete space-time point to another. We indicate the propagation along the temporal dimension from time step $s$ to time step $\tau$ with subscripts $s \to \tau$. Similarly, superscripts $j \to i$ indicate messages sent from the $j$-th node to the $i$-th. To avoid overloading the notation, we omit the layer superscript in the following. The message $\boldsymbol{r}_{s \to \tau}^{j \to i} \in \mathbb{R}^{d_h}$ from the $j$-th node at time step $s$ to the $i$-th node at time step $\tau$ is computed as

$$\boldsymbol{r}_{s \to \tau}^{j \to i} = \text{MLP}(\boldsymbol{h}_s^j, \boldsymbol{h}_\tau^i), \tag{5}$$

where, following the adopted terminology, target point representation $\boldsymbol{h}_\tau^i$ acts as query, source point representation $\boldsymbol{h}_s^j$ as key and the computed message $\boldsymbol{r}_{t \to \tau}^{j \to i}$ is the value. Notice that, differently from

the dot-product attention mechanism by Vaswani et al. [50], we take into account both source and target representations to compute the value, similarly to the Bahdanau attention [51]. To account for spatial information, this mechanism is used to perform an *inter-node* temporal cross-attention. More precisely, for every neighbor $j \in \mathcal{N}(i)$, we use $\boldsymbol{h}_\tau^i$ to query – separately – every encoding in $\boldsymbol{h}_{t:t+T}^j$ associated with a valid observation, and collect the messages in the set

$$\mathcal{R}_\tau^{j \to i} = \{ \boldsymbol{r}_{s \to \tau}^{j \to i} \mid \langle \boldsymbol{x}_s^j, \boldsymbol{q}_s^j \rangle \in \mathcal{X}_{t:t+T} \}. \tag{6}$$

Then, we linearly transform messages in $\mathcal{R}_\tau^{j \to i}$ (using trainable weights $\boldsymbol{W}_\alpha \in \mathbb{R}^{1 \times d_h}$) and obtain *message scores* $\alpha_{s \to \tau}^{j \to i}$ with a softmax layer, i.e.,

$$\alpha_{s \to \tau}^{j \to i} = \frac{\exp(\boldsymbol{W}_\alpha \boldsymbol{r}_{s \to \tau}^{j \to i})}{\sum_{\boldsymbol{r} \in \mathcal{R}_\tau^{j \to i}} \exp(\boldsymbol{W}_\alpha \boldsymbol{r})} \ . \tag{7}$$

Then, we aggregate temporal messages coming from each $j$-th node separately, and obtain an edge-level context vector $\boldsymbol{e}_\tau^{j \to i}$

$$\boldsymbol{e}_\tau^{j \to i} = \sum_{s:\, \boldsymbol{r}_{s \to \tau}^{j \to i} \in \mathcal{R}_\tau^{j \to i}} \alpha_{s \to \tau}^{j \to i} \cdot \boldsymbol{r}_{s \to \tau}^{j \to i}, \tag{8}$$

which encodes the observed sequence at each $j$-th node w.r.t. the $i$-th node and time step $\tau$. Analogously, to account for the observed sequence w.r.t. the $i$-th node itself, we exploit an *intra-node* temporal self-attention mechanism to query the encodings $\boldsymbol{h}_{t:t+T}^i$ corresponding to valid observations. Then, the resulting representations are weighted and aggregated to obtain a temporal context vector $\boldsymbol{c}_\tau^i$ as

$$\boldsymbol{r}_{s \to \tau}^i = \mathrm{MLP}\big(\boldsymbol{h}_s^i, \boldsymbol{h}_\tau^i\big) \tag{9}$$

$$\mathcal{R}_\tau^i = \{ \boldsymbol{r}_{s \to \tau}^i \mid \langle \boldsymbol{x}_s^i, \boldsymbol{q}_s^i \rangle \in \mathcal{X}_{t:t+T} \} \tag{10}$$

$$\boldsymbol{c}_\tau^i = \sum_{s:\, \boldsymbol{r}_{s \to \tau}^i \in \mathcal{R}_\tau^i} \alpha_{s \to \tau}^i \cdot \boldsymbol{r}_{s \to \tau}^i \tag{11}$$

where message score $\alpha_{s \to \tau}^i$ is obtained similarly as Eq. (7). Note that parameters are not shared between the cross-attention and self-attention blocks. After having obtained context vectors $\boldsymbol{c}_{\tau,(l)}^i$ and $\boldsymbol{e}_\tau^{j \to i,(l)}$, the encoding $\boldsymbol{h}_\tau^{i,(l)}$ is updated with a final aggregation step as

$$\boldsymbol{h}_\tau^{i,(l+1)} = \mathrm{MLP}\Big(\boldsymbol{h}_\tau^{i,(l)},\ \boldsymbol{c}_\tau^{i,(l)},\ \sum_{j \in \mathcal{N}(i)} \boldsymbol{e}_\tau^{j \to i,(l)}\Big). \tag{12}$$

After $L$ layers, we obtain imputations for all spatiotemporal points in $\mathcal{Y}_{t:t+T}$ with a nonlinear readout

$$\widehat{\mathcal{Y}}_{t:t+T} = \{ \hat{\boldsymbol{x}}_\tau^i = \mathrm{MLP}\big(\boldsymbol{h}_\tau^{i,(L)}\big) \mid \boldsymbol{q}_\tau^i \in \mathcal{Y}_{t:t+T} \}. \tag{13}$$

**Two-phase propagation** Masking out tokens in the target set allows SPIN to propagate only valid information. As a downside, this results in blocking the flow of information on paths through points in the target set. This can be problematic when the input observations are extremely sparse. Nonetheless, it is reasonable to assume that, after only a few propagation steps, the available information has already been partially diffused to locations with missing observations. At this point, interrupted paths can be unlocked, allowing for reaching higher-order neighborhoods. In practice, we introduce a hyperparameter $\eta$ to control the number of layers with masked connections and effectively split the propagation process into two phases. It is important to notice that what is being propagated in the second phase are learned representations, not observations (unavailable for masked tokens).

**Graph subsampling and hierarchical attention** Roughly speaking, the proposed spatiotemporal attention mechanism can be viewed as performing attention over the spatiotemporal graph $\mathcal{S}$, obtained by considering the product graph between space and time dimensions – with some connections pruned w.r.t. unavailable data. Let $N_{\max}, E_{\max}$ be the largest number of nodes and edges, respectively, among graphs in $\mathcal{G}_{t:t+T}$. Performing graph attention on the surrogate graph $\mathcal{S}$ has time and memory complexities that scale with $\mathcal{O}\big((N_{\max} + E_{\max})T^2\big)$ (sparse implementations can alleviate this complexity by replacing $T^2$ with the number of valid time step pairs). To reduce this computational burden – which undermines the application of the proposed method to large graphs and long time

horizons – we propose two different approaches. The straightforward approach consists in training the model by exploiting graph subsampling, using one of the many possible subsampling strategies from the literature (e.g., [52]). In practice, at training time, we sample a $k$-hop subgraph centered on $n$ target nodes and then compute the loss only w.r.t. these $n$ nodes. In this way, we can reduce the amount of computation required by acting on $n$ and $k$. This type of subsampling can also be seen as a form of regularization [53].

A more interesting and orthogonal approach to reduce complexity is to rewire the attention mechanism to be hierarchical [54]. We do this by adding $K$ dummy nodes that act as hubs for propagating information. Let $\boldsymbol{Z}^i \in \mathbb{R}^{K \times d_z}$ be the hub nodes' representations for central node $i$, and then, for hub $k$ proceed as follows.

1. Update $\boldsymbol{z}_k^i$ by querying $\{\boldsymbol{h}_\tau^i \mid \langle \boldsymbol{x}_\tau^i, \boldsymbol{q}_\tau^i \rangle \in \mathcal{X}_{t:t+T}\}$, i.e., node encodings associated with valid observations, obtaining $\tilde{\boldsymbol{z}}_k^i$;

2. Update node encoding $\boldsymbol{h}_\tau^i$ by querying updated $\widetilde{\boldsymbol{Z}}^i$ and $\widetilde{\boldsymbol{Z}}^j$ of every $j$-th neighbor in $\mathcal{N}(i)$.

The spatiotemporal attention is effectively split into two phases. At first, we update each hub node representation similarly as Eq. (9-11):

$$\boldsymbol{r}_{\tau,k}^i = \mathrm{MLP}\big(\boldsymbol{h}_\tau^i, \boldsymbol{z}_k^i\big) \qquad (14) \qquad\qquad \boldsymbol{c}_k^i = \sum_{\tau:\, \boldsymbol{r}_{\tau,k}^i \in \mathcal{R}_k^i} \alpha_{\tau,k}^i \cdot \boldsymbol{r}_{\tau,k}^i \qquad (16)$$

$$\mathcal{R}_k^i = \{\boldsymbol{r}_{\tau,k}^i \mid \langle \boldsymbol{x}_\tau^i, \boldsymbol{q}_\tau^i \rangle \in \mathcal{X}_{t:t+T}\} \qquad (15) \qquad\qquad \tilde{\boldsymbol{z}}_k^i = \mathrm{MLP}\big(\boldsymbol{z}_k^i, \boldsymbol{c}_k^i\big) \qquad (17)$$

Then, we obtain context vectors from the updated hub representations as:

$$\boldsymbol{r}_{k,\tau}^i = \mathrm{MLP}\big(\tilde{\boldsymbol{z}}_k^i, \boldsymbol{h}_\tau^i\big) \qquad (18) \qquad\qquad \boldsymbol{r}_{k,\tau}^{j\to i} = \mathrm{MLP}\big(\tilde{\boldsymbol{z}}_k^j, \boldsymbol{h}_\tau^i\big) \qquad (20)$$

$$\boldsymbol{c}_\tau^i = \sum_k \alpha_{k,\tau}^i \cdot \boldsymbol{r}_{k,\tau}^i \qquad (19) \qquad\qquad \boldsymbol{e}_\tau^{j\to i} = \sum_k \alpha_{k,\tau}^{j\to i} \cdot \boldsymbol{r}_{k,\tau}^{j\to i} \qquad (21)$$

and update node representation $\boldsymbol{h}_\tau^i$ as in Eq. (12). In this way, we can reduce the spatiotemporal attention complexity to $\mathcal{O}\left((N_{\max} + E_{\max})KT\right)$ with $K \ll T$, at the cost of introducing an information bottleneck. We initialize the hub representations at layer $l = 0$ with random trainable parameters.

**Spatiotemporal positional encoding**  We now come back to the problem of learning the spatiotemporal positional encodings $\boldsymbol{Q}_{t:t+T}$. The encoding has to capture both spatial and temporal structure and be decoupled from the observations $\boldsymbol{x}_t^i$ to allow inference for points in the target set. These encodings can be directly extracted from available exogenous information (e.g., sensor location and date/time features) or learned end-to-end jointly with the other model parameters. We propose a spatiotemporal positional encoding $\boldsymbol{q}_t^i = \rho(\boldsymbol{u}_t, \boldsymbol{v}^i)$ obtained by combining a *temporal encoding* $\boldsymbol{U} \in \mathbb{R}^{T \times d_u}$ and a *spatial encoding* $\boldsymbol{V} \in \mathbb{R}^{N \times d_v}$ with a non linear transformation $\rho$, e.g.,

$$\boldsymbol{q}_t^i = \mathrm{MLP}\big(\boldsymbol{u}_t, \boldsymbol{v}^i\big). \qquad (22)$$

For the temporal encoding $\boldsymbol{u}_t$, we use sine and cosine transforms of the time step $t$ w.r.t. a period of interest (e.g., day and/or week), to account for seasonalities. For the spatial encoding $\boldsymbol{v}^i$, we resort to a vector of learnable parameters different for each $i$-th node. More advanced methods could be considered [55, 56].

## 5  Empirical evaluation

In this section, we evaluate our method on three real-world datasets and compare the performance against state-of-the-art methods and standard approaches for MTSI. As the objective of our approach is to address the imputation problem in highly sparse settings, in a second experiment we assess how performance changes as the percentage of missing values increases.

### 5.1  Experimental setting

In the following experiments, we consider both **SPIN** and the hierarchical version **SPIN-H** (Sec. 4). The figure of merit used is the *mean absolute error* (MAE), averaged across imputation windows.

Table 1: Performance (in terms of MAE) averaged over 5 independent runs.

| | Block missing | | Point missing | | Simulated failures | |
|---|---|---|---|---|---|---|
| | PEMS-BAY | METR-LA | PEMS-BAY | METR-LA | AQI-36 | AQI |
| Mean | $5.46 \pm 0.00$ | $7.48 \pm 0.00$ | $5.42 \pm 0.00$ | $7.56 \pm 0.00$ | $53.48 \pm 0.00$ | $39.60 \pm 0.00$ |
| KNN | $4.30 \pm 0.00$ | $7.79 \pm 0.00$ | $4.30 \pm 0.00$ | $7.88 \pm 0.00$ | $30.21 \pm 0.00$ | $34.10 \pm 0.00$ |
| MF | $3.28 \pm 0.01$ | $5.46 \pm 0.02$ | $3.29 \pm 0.01$ | $5.56 \pm 0.03$ | $30.54 \pm 0.26$ | $26.74 \pm 0.24$ |
| MICE | $2.94 \pm 0.02$ | $4.22 \pm 0.05$ | $3.09 \pm 0.02$ | $4.42 \pm 0.07$ | $30.37 \pm 0.09$ | $26.98 \pm 0.10$ |
| VAR | $2.09 \pm 0.10$ | $3.11 \pm 0.08$ | $1.30 \pm 0.00$ | $2.69 \pm 0.00$ | $15.64 \pm 0.08$ | $22.95 \pm 0.30$ |
| rGAIN | $2.18 \pm 0.01$ | $2.90 \pm 0.01$ | $1.88 \pm 0.02$ | $2.83 \pm 0.01$ | $15.37 \pm 0.26$ | $21.78 \pm 0.50$ |
| BRITS | $1.70 \pm 0.01$ | $2.34 \pm 0.01$ | $1.47 \pm 0.00$ | $2.34 \pm 0.00$ | $14.50 \pm 0.35$ | $20.21 \pm 0.22$ |
| SAITS | $1.56 \pm 0.01$ | $2.30 \pm 0.01$ | $1.40 \pm 0.03$ | $2.26 \pm 0.00$ | $18.16 \pm 0.42$ | $21.33 \pm 0.15$ |
| Transformer | $1.70 \pm 0.02$ | $3.54 \pm 0.00$ | $0.74 \pm 0.00$ | $2.16 \pm 0.00$ | $11.98 \pm 0.53$ | $18.11 \pm 0.25$ |
| GRIN | $1.14 \pm 0.01$ | $2.03 \pm 0.00$ | $\mathbf{0.67} \pm \mathbf{0.00}$ | $\mathbf{1.91} \pm \mathbf{0.00}$ | $12.08 \pm 0.47$ | $14.73 \pm 0.15$ |
| **SPIN** | $\mathbf{1.06} \pm \mathbf{0.02}$ | $\mathbf{1.98} \pm \mathbf{0.01}$ | $0.70 \pm 0.01$ | $\mathbf{1.90} \pm \mathbf{0.01}$ | $11.77 \pm 0.54$ | $\mathbf{13.92} \pm \mathbf{0.15}$ |
| **SPIN-H** | $\mathbf{1.05} \pm \mathbf{0.01}$ | $2.05 \pm 0.02$ | $0.73 \pm 0.01$ | $1.96 \pm 0.03$ | $\mathbf{10.89} \pm \mathbf{0.27}$ | $14.41 \pm 0.13$ |

We consider only the *out-of-sample* scenario [4], in which every parametric model is trained and tested on disjoint sets. All the baselines have been implemented in PyTorch [57] using the Torch Spatiotemporal library[2] [58] and, whenever possible, open-source code provided by the authors. The code to reproduce the experiments of the paper is available online[3]. Please refer to the appendix for more details about the experimental setup.

**Datasets**   We consider three openly available datasets coming from real-world SNs. The first two, namely **PEMS-BAY** and **METR-LA** [2], are two widely used benchmarks in spatiotemporal forecasting literature. Each of them records traffic measurements every 5 minutes from 325 speed sensors in San Francisco Bay Area and 207 in Los Angeles County Highway, respectively. Since the original datasets have a low number of missing values, we use the same setup of [4] to inject missing data with two different policies: 1) *Block missing*, in which we randomly mask out $5\%$ of the available data and, in addition, we simulate a failure lasting for $S \sim \mathcal{U}(12, 48)$ steps with $0.15\%$ probability; 2) *Point missing*, in which we randomly drop $25\%$ of the available data. As a third dataset, we consider **AQI** [59], which collects one year of hourly measurements of air pollutants from 437 air quality monitoring stations over 43 cities in China. We consider also a smaller version of this dataset (**AQI-36**) with only the 36 sensors scattered over the city of Beijing. This dataset is a popular benchmark for imputation for the high number of missing values ($25.67\%$ in the complete dataset) and provides a mask for evaluation that simulates the true missing data distribution [15]. For a given month, such a mask replicates the missing values patterns of the previous month, making this scenario more similar to the *Block missing* setting, for a total of $\approx 36\%$ missing data. In all settings, all the valid observations masked out are used as targets for evaluation. We obtain an adjacency matrix from the pairwise distance of sensors following previous works [2–4].

**Baselines**   As the target of our approach is the processing of spatiotemporal graphs with missing observations, we compare our method against GRIN [4], a graph-based bidirectional RNN for MTSI with state-of-the-art performance. We then consider a spatiotemporal Transformer, where we alternate temporal and spatial Transformer encoder layers from [50] and replace missing values with a [MASK] token (as in [12]). We consider also other deep imputation methods: 1) SAITS [25], a recent attention-based architecture based on diagonally-masked self-attention; 2) BRITS [6], which leverages on a bidirectional RNN; 3) rGAIN, an adversarial approach which shares similarities with GAIN [23] and SSGAN [21]. Finally, we report results of simpler methods that impute missing values using 4) node-level sequence mean (MEAN) or 5) neighbors mean (KNN); 6) Matrix Factorization (MF); 7) MICE [60]; 8) VAR, a vector autoregressive one-step-ahead predictor. Whenever possible, we use results from [4].

---

[2]https://github.com/TorchSpatiotemporal/tsl
[3]https://github.com/Graph-Machine-Learning-Group/spin

Table 2: Performance (MAE) with increasing data sparsity in the *Point missing* setting (averaged over 5 evaluation masks).

| | METR-LA | | | PEMS-BAY | | | AQI | | |
|---|---|---|---|---|---|---|---|---|---|
| | Missing rate | | | Missing rate | | | Missing rate | | |
| | 50 % | 75 % | 95 % | 50 % | 75 % | 95 % | 50 % | 75 % | 95 % |
| BRITS | $2.52 \pm 0.00$ | $3.02 \pm 0.00$ | $5.19 \pm 0.02$ | $1.55 \pm 0.00$ | $2.17 \pm 0.00$ | $3.91 \pm 0.02$ | $14.90 \pm 0.03$ | $18.29 \pm 0.03$ | $29.83 \pm 0.07$ |
| SAITS | $2.48 \pm 0.00$ | $3.74 \pm 0.01$ | $6.72 \pm 0.01$ | $1.50 \pm 0.00$ | $2.96 \pm 0.01$ | $7.40 \pm 0.01$ | $15.36 \pm 0.02$ | $20.64 \pm 0.05$ | $34.57 \pm 0.05$ |
| Transformer | $2.31 \pm 0.00$ | $2.71 \pm 0.00$ | $5.13 \pm 0.01$ | $0.85 \pm 0.00$ | $1.13 \pm 0.00$ | $2.70 \pm 0.01$ | $9.11 \pm 0.02$ | $12.56 \pm 0.05$ | $25.65 \pm 0.11$ |
| GRIN | $2.05 \pm 0.00$ | $2.39 \pm 0.00$ | $4.08 \pm 0.02$ | $\mathbf{0.79 \pm 0.00}$ | $1.09 \pm 0.00$ | $2.70 \pm 0.01$ | $8.43 \pm 0.01$ | $10.97 \pm 0.02$ | $20.38 \pm 0.10$ |
| **SPIN** | $2.02 \pm 0.00$ | $2.24 \pm 0.00$ | $2.89 \pm 0.01$ | $\mathbf{0.79 \pm 0.00}$ | $1.00 \pm 0.00$ | $1.71 \pm 0.00$ | $\mathbf{8.15 \pm 0.01}$ | $\mathbf{9.96 \pm 0.02}$ | $\mathbf{15.51 \pm 0.08}$ |
| **SPIN-H** | $\mathbf{2.01 \pm 0.00}$ | $\mathbf{2.20 \pm 0.00}$ | $\mathbf{2.82 \pm 0.00}$ | $\mathbf{0.79 \pm 0.00}$ | $\mathbf{0.97 \pm 0.00}$ | $\mathbf{1.68 \pm 0.00}$ | $8.67 \pm 0.02$ | $10.27 \pm 0.02$ | $15.75 \pm 0.07$ |

Table 3: Performance (MAE) with an increasing number of simulated failures in the *Block missing* setting (averaged over 5 evaluation masks).

| | METR-LA | | | PEMS-BAY | | | AQI | | |
|---|---|---|---|---|---|---|---|---|---|
| | Failure probability | | | Failure probability | | | Failure probability | | |
| | 5 % | 10 % | 15 % | 5 % | 10 % | 15 % | 5 % | 10 % | 15 % |
| BRITS | $5.87 \pm 0.03$ | $7.26 \pm 0.06$ | $8.29 \pm 0.07$ | $4.14 \pm 0.05$ | $5.41 \pm 0.08$ | $5.84 \pm 0.04$ | $24.09 \pm 0.30$ | $31.90 \pm 0.26$ | $37.62 \pm 0.42$ |
| SAITS | $4.73 \pm 0.07$ | $6.66 \pm 0.05$ | $7.27 \pm 0.03$ | $3.88 \pm 0.09$ | $7.62 \pm 0.21$ | $8.01 \pm 0.11$ | $20.78 \pm 0.30$ | $30.16 \pm 0.39$ | $36.34 \pm 0.33$ |
| Transformer | $6.03 \pm 0.04$ | $7.19 \pm 0.05$ | $8.06 \pm 0.05$ | $3.69 \pm 0.06$ | $5.09 \pm 0.05$ | $6.02 \pm 0.04$ | $29.21 \pm 0.33$ | $33.62 \pm 0.16$ | $37.31 \pm 0.14$ |
| GRIN | $3.05 \pm 0.02$ | $4.52 \pm 0.05$ | $5.82 \pm 0.06$ | $2.26 \pm 0.03$ | $3.45 \pm 0.06$ | $4.35 \pm 0.04$ | $15.62 \pm 0.24$ | $22.08 \pm 0.39$ | $29.03 \pm 0.42$ |
| **SPIN** | $2.71 \pm 0.02$ | $3.32 \pm 0.02$ | $3.87 \pm 0.05$ | $\mathbf{1.78 \pm 0.03}$ | $\mathbf{2.15 \pm 0.03}$ | $\mathbf{2.41 \pm 0.02}$ | $\mathbf{14.29 \pm 0.24}$ | $\mathbf{18.71 \pm 0.34}$ | $\mathbf{24.34 \pm 0.46}$ |
| **SPIN-H** | $\mathbf{2.64 \pm 0.02}$ | $\mathbf{3.17 \pm 0.02}$ | $\mathbf{3.61 \pm 0.04}$ | $\mathbf{1.75 \pm 0.04}$ | $2.16 \pm 0.03$ | $2.48 \pm 0.02$ | $14.55 \pm 0.26$ | $19.37 \pm 0.36$ | $25.38 \pm 0.37$ |

**Computational complexity** We recall that time and memory complexity of SPIN and SPIN-H scales with $\mathcal{O}\left((N_{\max} + E_{\max})T^2\right)$ and $\mathcal{O}\left((N_{\max} + E_{\max})KT\right)$, respectively. For the sake of comparison, here we also report the asymptotic complexity for the spatiotemporal Transformer and GRIN. The Transformer alternates temporal attention – $\mathcal{O}\left(N_{\max}T^2\right)$ – and spatial attention – $\mathcal{O}\left(TN_{\max}^2\right)$ – with a resulting $\mathcal{O}\left((N_{\max} + T)N_{\max}T\right)$ complexity. As pertaining to GRIN, let $R$ be the spatial receptive field (i.e., number of graph convolution layers) of the inner MPGRU cell. Then, GRIN's time complexity scales with $\mathcal{O}\left(TRE_{\max}\right)$ in the unidirectional model. Note that while most of the operations in the attention-based models can be executed in parallel, GRIN needs to process the entire sequence sequentially, with a consequent slowdown.

## 5.2 Results

Table 1 reports experimental results. Both SPIN methods outperform the baselines in almost all scenarios. Not surprisingly, improvements are more evident when entire blocks of data are missing, as in the AQI datasets and *Block missing* settings. Differently from the autoregressive methods, the sparse spatiotemporal attention mechanism of SPIN, in fact, allows for propagating far-apart observations without the need for intermediate – and biased – predictions. Conversely, in the *Point missing* setting, SPIN methods perform on par with the state of the art. With respect to the spatiotemporal Transformer, SPIN performs better in all settings except for AQI-36, which can be attributed to the ineffectiveness of spatial attention alone in determining the dependencies among the different spatial locations. Notice also that, in almost all cases, SPIN-H performs on par with SPIN (even better in the smaller dataset AQI-36), making it a valid lightweight alternative to SPIN.

To further assess the advantages of SPIN in reconstructing sequences with highly sparse observations, we compare our methods against the most relevant baselines on the same datasets, but considering two additional scenarios with increased sparsity. In the first setting, the missing rate is progressively increased by associating to each observation an increasing probability $p$ of being removed; this corresponds to a sparser version of the *Point missing* scenario of the previous experiment. In the second case, we instead operate in the *Block missing* setting by increasing the probability $\bar{p}$ of a failure at each step, i.e., the probability for each sensor of going offline for a random number $S \sim \mathcal{U}(12, 36)$ of future (consecutive) time steps. In practice, we test the models on the same test split of the previous experiment but, instead of using the previous evaluation masks, we update them according to the different missing data distributions. Note that higher missing rates and failure probabilities correspond to higher numbers of (consecutive) missing values. In the *Block missing* case,

failure probabilities $\bar{p} = 5\%$, $\bar{p} = 10\%$, and $\bar{p} = 15\%$ correspond to a missing rate of $\approx$ 70-75%, $\approx$ 90-92%, and $\approx$ 96-97%, respectively. For the traffic datasets, we use the weights of the models trained on the *Point missing* setting of Table 1. Table 2 and Table 3 show results (in terms of MAE) averaged over 5 different evaluation masks for both scenarios and across different missing rates and failure probabilities. In all the considered experiments, SPIN-based models rank as the best-performing methods for any sparsity level. Moreover, our approach is robust to changes in the missing data distribution. In fact, compared to the baselines, the performance of both SPIN and SPIN-H deteriorates at a slower rate as the sparsity data increases. Notably, GRIN would require much more data to match the performance of SPIN, and the improvement over other attention-based methods, i.e, SAITS and Transformer, is also striking.

## 6    Conclusion and Future Works

We introduced a graph-based architecture, named SPIN, to reconstruct missing observations in sparse spatiotemporal time series. To overcome the major limitations of autoregressive methods, we designed a novel sparse spatiotemporal attention mechanism to propagate valid observations through discrete points in time and space, jointly. Furthermore, we showed how the time and space complexities of the approach can be drastically reduced by considering a novel hierarchical attention mechanism. Empirical analysis shows that the proposed method outperforms by a wide margin state-of-the-art methods for imputation in highly sparse settings. We noticed that in some extremely sparse *training* settings, SPIN might suffer from a lack of supervision that may slow down learning; future works might try to address this problem by introducing an additional auxiliary learning task. Other possible extensions could investigate spatiotemporal positional encoding methods and their application in SNs.

## Acknowledgements

This work was supported by the Swiss National Science Foundation project FNS 204061: *Higher-Order Relations and Dynamics in Graph Neural Networks*. The authors wish to thank the Institute of Computational Science at USI for granting access to computational resources.

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
