# OpenReview forum: "Learning to Reconstruct Missing Data from Spatiotemporal Graphs with Sparse Observations"
_NeurIPS.cc/2022/Conference — NeurIPS 2022 Accept_

### Official Review · Reviewer_K4Xt · 2022-07-05

**Rating:** 6
**Confidence:** 2
**Soundness:** 3 good
**Presentation:** 3 good
**Contribution:** 2 fair

**Summary:**

This paper introduces SPIN, a spatiotemporal graph network with a new spatiotemporal attention for reconstructing missing data points in multivariate time series. The proposed attention mechanism consists of the inter-node cross-attention and the intra-node self-attention. Compared to the autoregressive methods propagating the biased representations, one advantage of SPIN is that the reconstruction only relies on the observations. They also propose the hierarchical attention to reduce the computational cost. Their empirical evaluation shows that SPIN outperforms existing baselines generally and is clearly advantageous when the missing ratio is large.

**Questions:**

1. What is $w_\alpha$ in equation (7)? Is it a learnable paramter?
2. Why do you use the Bahdanau attention instead of the dot-product attention?

**Limitations:**

I can find one sentence in the conclusion discussing the limitation. But I don't find any discussion about the potential societal impact.

**Strengths And Weaknesses:**

Strengths:

* The paper tackles an important problem and the proposed spatiotemporal attention mechanism is sound.
* The visualizations of the model architecture and the attention mechanism are illustrative and help in understanding.
* The method section is well structured and easy to follow even though there are many design details.
* The related work section offers a good overview of the existing work.

Weaknesses:

1. Although the quantitative results show the effectiveness of the method, it would be even better if there are qualitative results visualizing either the imputed data points or the cross-attention weights and self-attention weights.
2. Although the authors compare the computation complexity between SPIN and SPIN-H, it would be more comprehensive if the comparison to baselines in terms of time cost or memory cost could be reported.

---

> ### Author Response · Authors · 2022-08-02
> **Response to Reviewer K4Xt comments**
>
> Thank you for your feedback and your valuable suggestions.
>
> > Although the quantitative results show the effectiveness of the method, it would be even better if there are qualitative results visualizing either the imputed data points or the cross-attention weights and self-attention weights.
>
> In the last revision (Appendix C), we added a figure showing the performance of SPIN-H on virtual sensing, i.e., reconstructing entire missing sequences on new nodes (nodes with no data during training). Unfortunately, visualizing the attention weights could be quite hard to interpret in deep models and large datasets, such as the ones considered in our work.
>
> > Although the authors compare the computation complexity between SPIN and SPIN-H, it would be more comprehensive if the comparison to baselines in terms of time cost or memory cost could be reported.
>
> We now added in Appendix A the asymptotic complexities of the most relevant baselines. There, we also report running times and hardware used for the training of SPIN-based models. Since we distributed the training across batches and different computing nodes, a direct comparison of the actual timings and memory costs is heavily influenced by implementation details.
>
> > 1. What is w_a in equation (7)? Is it a learnable parameter?
>
> Yes, it is the column vector with the learnable weights of the message transformation. Thank you for pointing this out, we made it more clear in the paper now.
>
> > 2. Why do you use the Bahdanau attention instead of the dot-product attention?
>
> Since we are performing attention across different nodes and representations instead of the usual self-attention, we opted for a more expressive mechanism to compute messages (i.e., MLPs). Note that this type of attention mechanism is often preferred in GNNs (e.g., see GAT [1]).
>
> [1] Veličković, Petar, et al. "Graph Attention Networks." International Conference on Learning Representations. 2018.

---

### Official Review · Reviewer_tTGX · 2022-07-09

**Rating:** 5
**Confidence:** 4
**Soundness:** 2 fair
**Presentation:** 3 good
**Contribution:** 2 fair

**Summary:**

In this paper, the authors propose an attention-based model for missing data reconstruction from spatiotemporal graphs. The core part of the method is a sparse spatiotemporal attention mechanism that learns representations localized in time and space from sparse data. Empirical results show that the proposed method outperforms existing approaches, especially when the missing rate is high.

**Questions:**

1. Line 56: “Conversely”. From my understanding of the proposed “spatiotemporal cross-attention” and “temporal self-attention”, they also set the attention coefficients of spatiotemporal points with missing values to zero. How is this method different from the so-called “trivial way”: constraining the attention coefficients of time steps corresponding to missing data to zero?
2. Line 58: “we seek to design an architecture where each processing stage is aligned with the task of reconstructing missing spatiotemporal observations.”. How is the design in each stage aligned with the task of reconstructing?
3. Line 60: “Our model can be seen as a learned, self-organizing, spatiotemporal diffusion process”. How is it related to the diffusion process?
4. Line 66: “reconstructing observations at arbitrary locations”. As described in lines 265-266, the spatial encoding is trained as learnable parameters different for each node. How can the proposed method reconstruct observations at arbitrary locations? What if the queried location has never appeared in training data and its embedding is never defined in training? The method actually assumes that the locations in evaluation are exactly the same locations in training.
5. Line 71: “ equipped with inductive biases that make the model tailored for the multivariate time series imputation task”. What is the inductive bias that makes the model tailored for the multivariate time series imputation task?


**Limitations:**

The authors have adequately addressed the limitations.

**Strengths And Weaknesses:**

Strengths:
1. The writing is clear and the figures are illustrative.
2. Experiments show promising results.

Weaknesses:
1. Some claims are questionable and might be over-claimed (see “Questions”).
2. There are already some works on sparse attention with spatiotemporal data. (For example, [1] has a similar idea of utilizing valid spatiotemporal data points only to generate representations. There might be more works along this line though.) The authors shall provide more context of sparse attention with spatiotemporal data.
3. The authors should have more ablation study experiments to explain the effect of the claimed contributions - sparse spatiotemporal attention mechanism, the inductive biases, etc.
4. In Figure 1, there should be an “MLP” block between the concatenation of (spatiotemporal cross-attention, temporal self-attention, skip connection) and “Add & Norm”, according to Equation 12.

[1]. Huang, Z., Sun, Y., & Wang, W. (2020). Learning continuous system dynamics from irregularly-sampled partial observations. Advances in Neural Information Processing Systems, 33, 16177-16187.

---

> ### Author Response · Authors · 2022-08-02
> **Response to Reviewer tTGX comments (part I)**
>
> Thank you for your comments and feedback, we will do our best to clarify your doubts about our work.
>
> > There are already some works on sparse attention with spatiotemporal data. (For example, [1] has a similar idea of utilizing valid spatiotemporal data points only to generate representations. There might be more works along this line though.) The authors shall provide more context of sparse attention with spatiotemporal data.
>
> We discuss attention-based models in the second paragraph of related works (Line 102-109). In particular, we discuss the use of attention in graph convolutional filters (L102-103) and spatiotemporal GNNs using attention (L103-106). We also discuss attention-based methods for multivariate time series imputation that do not take explicitly into account relational dependencies (L92-95). Nonetheless, the suggested work is indeed a relevant reference, we included it in the last revision of the paper.
>
> > The authors should have more ablation study experiments to explain the effect of the claimed contributions - sparse spatiotemporal attention mechanism, the inductive biases, etc.
>
> We included an ablation study in the last revision (Appendix D). More specifically, it investigates the contribution of the intra-node self-attention and inter-node cross-attention mechanisms to imputation accuracy. The Transformer variant can also be seen as an ablation of the core components of SPIN (e.g., the spatiotemporal attention mechanism). As such, it provides a comparison between our method and a more standard attention-based architecture.
>
> > In Figure 1, there should be an “MLP” block between the concatenation of (spatiotemporal cross-attention, temporal self-attention, skip connection) and “Add & Norm”, according to Equation 12.
>
> We initially excluded it from the figure to focus on the most relevant blocks. After your request, we updated the figures accordingly. Thank you for pointing this out.
>
> > 1. Line 56: “Conversely”. From my understanding of the proposed “spatiotemporal cross-attention” and “temporal self-attention”, they also set the attention coefficients of spatiotemporal points with missing values to zero. How is this method different from the so-called “trivial way”: constraining the attention coefficients of time steps corresponding to missing data to zero?
>
> “Masking” here happens at a more structural level compared to the simple zeroing out of the attention coefficients in the first layer. First of all, we handle the observed and target set differently throughout the architecture, starting from the initial encoding (see Eq. 4) to the propagation of information at each step. Furthermore, information flow is blocked for several propagation steps (not only the first one), differently from other attention-based imputation architectures [1]. Note that, in standard architectures, simply masking out raw observations does not prevent information flow from observed to target set if the number of layers is $L \geq 2$. Indeed, even masking out $h_t^i$ at layer $l$, there is a path to $h_t^i$ at layer $l-1$ by reading from neighboring nodes $(h_t^j, j \in \mathcal{N}(i))$ at layer $l$.
>
> > 2. Line 58: “we seek to design an architecture where each processing stage is aligned with the task of reconstructing missing spatiotemporal observations.”. How is the design in each stage aligned with the task of reconstructing?
>
> As we specified at the beginning of Sec. 4, we model all the spatiotemporal dependencies ($p(x_t^i|X_{<t})$, …, $p(x_t^i|\{x_t^{j \neq i}})$)  with an architecture that explicitly learns to weigh them and propagate information across the spatiotemporal structure (i.e., without the limits of autoressive models). At every step of the method, we update the representation of the target spatiotemporal point to be imputed by relying only on the acquired (valid) observations at the node and its neighbors. This allows, for instance, for getting rid of intermediate predictions/imputations needed in autoregressive approaches, thus enabling us to model Eq. 3.
>
> > 3. Line 60: “Our model can be seen as a learned, self-organizing, spatiotemporal diffusion process”. How is it related to the diffusion process?
>
> Here we use the term “diffusion” to indicate the incremental (learned) spatiotemporal propagation of information that happens at each layer. We substituted “diffusion” with “propagation” to avoid ambiguity.

---

> ### Author Response · Authors · 2022-08-02
> **Response to Reviewer tTGX comments (part II)**
>
> > 4. Line 66: “reconstructing observations at arbitrary locations”. As described in lines 265-266, the spatial encoding is trained as learnable parameters different for each node. How can the proposed method reconstruct observations at arbitrary locations? What if the queried location has never appeared in training data and its embedding is never defined in training? The method actually assumes that the locations in evaluation are exactly the same locations in training.
>
> It is possible to add a virtual node in the graph at training phase (with no observations available) and learn an embedding also for this fictitious node (a gradient would be propagated from the loss computed at the other nodes). This would be similar to the virtual sensor experiment in [2]. As one may expect, performance at virtual sensors would be sub-par w.r.t. real sensors with observations during training. We included further considerations on virtual sensing in the last revision, Appendix C.
> Moreover, it is possible to use sensor metadata (e.g., geographic coordinates) as spatial positional encodings instead of free parameters as node embeddings.
>
> Nonetheless, “arbitrary locations” can be misinterpreted also given these justifications, therefore we rephrased the sentence in the last revision.
>
> > 5. Line 71: “ equipped with inductive biases that make the model tailored for the multivariate time series imputation task”. What is the inductive bias that makes the model tailored for the multivariate time series imputation task?
>
> We partly answered some of these doubts in our answer to Question 2. Here, by inductive biases we mean (i) the relational inductive biases (modeled by the graph edges), (ii) the spatiotemporal propagation of information from the observed to the target set. We refer the reviewer to the first paragraph of Sec.4 which contains a high-level description of what we believe are the properties that make a model aligned with the imputation task.
>
>
> [1] Du, Wenjie, David Côté, and Yan Liu. "SAITS: Self-Attention-based Imputation for Time Series." arXiv preprint arXiv:2202.08516 (2022).
>
> [2] Cini, Andrea, Ivan Marisca, and Cesare Alippi. "Filling the G_ap_s: Multivariate Time Series Imputation by Graph Neural Networks." International Conference on Learning Representations. 2021.

---

> ### Comment · Reviewer_tTGX · 2022-08-08
> **Thanks for the authors' clarification**
>
> I appreciate the authors' clarification and responses to my concerns. While I am still in doubt whether the submission will have a high impact, I think the authors' responses have at least clarified most of the ambiguous expressions in the main text. I have raised my rating to 5.

---

### Official Review · Reviewer_PQtk · 2022-07-11

**Rating:** 6
**Confidence:** 2
**Soundness:** 3 good
**Presentation:** 4 excellent
**Contribution:** 3 good

**Summary:**

The paper proposes two variants (a basic one, SPIN, and a hierarchical one, SPIN-H) of attention-based message-passing networks to learn representations of a multivariate time series over a static graph, and which cope well with a large amount of missing data. The network learns representations that can reconstruct missing observations not only from the other temporal data at the node, but also from its neighboring nodes. Its experimental performance is better than other state-of-the art methods when a long sequence of temporal data is missing ("block missing") or when the amount of missing data is large, and comparable to the best methods when data are removed i.i.d in time ("point missing") in a moderate amount (<= 50%).



**Questions:**

1) SPIN is designed to handle blocks of missing data at one point over a long period of time. In Table 2, the amount of missing data gets very large, but are the data removed i.i.d (as in the point missing model of Table 1) or not (e.g., by picking some nodes which lose a lot of data?)

2) SPIN-H is computationally less expensive than SPIN by a K/T factor, but Appendix A shows that the number of parameters is 10 fold larger. How do you compare the two in that regard? How do you assess the cost of introducing an information bottleneck mentioned at L255?

3) Why does SPIN-H work better than SPIN on AQI-36, and worse on AQI (Table 1)?

4) Why does SPIN-H work better on the two other datasets PEMS-BAY and METR-LA when the amount of missing data is >= 50% (Table 2) and worse when the  the amount of missing data is = 25% (Table 1)? Does the hierarchy provide some form of regularization, like graph sub-sampling, which large amounts of missing data?

5) In eq (7), how is w_\alpha obtained - directly from (5)?

6) Minor nits: What does the green surface on the left of Fig 2 represent?
Typos at L119 (Pearson and not Person), L161 (adds and not add), L206 (to weigh and not to weight).

-----------------------------------------------------------

Update after discussion with authors: The rebuttal addresses (most of) my comments. The new results of Appendix E strengthen the interest of SPIN(-H) over other benchmarks. I still think that the paper would benefit from a deeper comparative analysis of the computational and memory complexities of SPIN vs SPIN-H, as well as vs the other benchmarks, and I understand from the authors' reply in the discussion that this will be done. I agree with the other reviewers that the paper impact might be limited, but overall I find the paper to be well written and bringing a contribution to the state of the art, hence my score for the review.


**Strengths And Weaknesses:**

The paper is well written, and the approach makes sense, in particular to cope with long blocks of missing data a particular node.

The performance of SPIN and SPIN-H is compared empirically to multiple benchmarks in Section 5, but only under the perspective of MAE performance. In particular, the computational and memory complexities are not compared to these methods, although the justification for SPIN-H is precisely a decrease in complexity by a factor K/T over SPIN, where K << T is the number of "hubs" added in the network and T the length of the time-sequence taken for learning.

The paper could investigate deeper other properties of SPIN and SPIN-H (see also below).

---

> ### Author Response · Authors · 2022-08-02
> **Response to Reviewer PQtk comments**
>
> Thank you for your insightful comments, below we reply to your questions and comments.
>
> > The paper proposes two variants (a basic one, SPIN, and a hierarchal one, SPIN-H) of attention-based message-passing networks to learn representations of a multivariate time series over a static graph,
>
> The graph can be also dynamic, i.e., we can use a different adjacency matrix at each time step, provided that nodes are identified throughout the time series.
>
> > The performance of SPIN and SPIN-H is compared empirically to multiple benchmarks in Section 5, but only under the perspective of MAE performance. In particular, the computational and memory complexities are not compared to these methods,
>
> We added a discussion about the computational complexities of the methods in Appendix A, where we also report timings for SPIN and SPIN-H.
>
> > 1. SPIN is designed to handle blocks of missing data at one point over a long period of time. In Table 2, the amount of missing data gets very large, but are the data removed i.i.d (as in the point missing model of Table 1) or not (e.g., by picking some nodes which lose a lot of data)?
>
> Your first guess is correct. In Tab. 2 we simulate i.i.d. missing data. However, in the last revision, we added an additional table in Appendix E with results referring to longer blocks of missing data (as in the block missing case). For this additional experiment, at each time step, we simulate a failure extended in time (from 12 to 36 time steps) with constant probability (that we chose as 0.05, 0.1, and 0.15). Results show that SPIN variants are the only models able to handle this scenario.
>
> > 2. SPIN-H is computationally less expensive than SPIN by a K/T factor, but Appendix A shows that the number of parameters is 10 fold larger.  How do you compare the two in that regard?
>
> The number of parameters for SPIN-H is higher w.r.t. SPIN not only for the additional weights needed to handle the hub nodes but also because in SPIN-H we used a deeper and wider network, enabled by the reduced computational cost.
>
> > How do you assess the cost of introducing an information bottleneck mentionned at L255?
>
> It is difficult to provide a general answer as the cost is application dependent. What can be said, is that the reduced memory complexity allows the designer to use wider and deeper networks, thus partially amortizing this cost.
>
> > 3. Why does SPIN-H work better than SPIN on AQI-36, and worse on AQI (Table 1)?
>
> It is difficult to provide a definitive answer. Our intuition is that in smaller datasets the use of hub nodes does not introduce a relevant bottleneck in the propagation process, while this might happen in larger graphs. Note also that the difference in accuracy between the two approaches is very limited.
>
> > 4. Why does SPIN-H work better on the two other datasets PEMS-BAY and METR-LA when the amount of missing data is >= 50% (Table 2) and worse when the the amount of missing data is = 25% (Table 1)? Does the hierarchy provide some form of regularization, like graph sub-sampling, which large amounts of missing data?
>
> Yes, it is reasonable to expect SPIN-H to introduce a positive regularization in the sparse setting. It is also expected that the non-hierarchical SPIN works better when there is more spatiotemporal information to propagate: this is consistent with our observations in the answer to your previous question.
>
> > 5. In eq (7), how is w_\alpha obtained - directly from (5)?
>
> In Eq. (7) we obtain message scores alpha with a softmax over a linear transformation of the message itself ($r^{j \rightarrow i}\_{s \rightarrow \tau}$) where $w\_{\alpha}$ is the column vector of the learnable weights. We clarified this in the last revision.
>
> > 6. Minor nits: What does the green surface on the left of Fig 2 represent? Typos at L119 (Pearson and not Person), L161 (adds and not add), L206 (to weigh and not to weight).
>
> The green box is just to highlight the target spatiotemporal point for imputation, we made it clearer in the caption now. Thank you for spotting the typos, we fixed them in the last revision of the paper.

---

> > ### Comment · Reviewer_PQtk · 2022-08-08
> > **Short feedback on rebuttal**
> >
> > Thanks for the rebuttal, which does address (most of) my comments.
> > - The new results of Appendix E strengthen the interest of SPIN(-H) over other benchmarks.
> > - I still think that the paper would benefit from a deeper comparative analysis (than a couple of lines (L255-257)) of the computational and memory complexities of SPIN vs SPIN-H, as well as vs the other benchmarks.

---

> > > ### Author Response · Authors · 2022-08-09
> > > **Response to Reviewer PQtk rebuttal feedback**
> > >
> > > We agree with the reviewer that the paper could be improved in this regard. In the next revision, we will integrate lines 600-608 of Appendix A (which reports a deeper analysis of the computational complexity of SPIN/SPIN-H w.r.t. the baselines) and improve the discussion on this aspect. We also point out that qualitative timings and memory requirements for SPIN and SPIN-H are also reported in lines 612-615, we will consider adding them to the main body of the paper.
> > >
> > > Thank you again for your valuable feedback.

---

### Official Review · Reviewer_PQKX · 2022-07-12

**Rating:** 7
**Confidence:** 3
**Soundness:** 4 excellent
**Presentation:** 4 excellent
**Contribution:** 3 good

**Summary:**

The paper presents a novel neural architecture for the imputation task in the context of multivariate spatio temporal series (for instances in sensor networks). The core idea is derived from the constatation that autoregressive models for this task are prone to error propagation, especially in the case of huge sparsity of available data. The authors propose to use an attention mechanism - spatial and temporal - in order to learn representations of the graph nodes and to use those embeddings to derivate the missing values rather than the values of previous/next temporal steps. The model is derived from the now classical GNN framework. A spatio-temporal cross-attention layer is proposed for the message passing between nodes and a temporal self-attention layer for the inference of a context representation vector of the time history of the considered node. Two approaches are presented to reduce the time complexity of the learning phase, the first one using classic subsampling methods in graphs, and the other one based on a hierarchical attention mechanism. Experiments are conducted on 4 datasets to compare the proposed approach to state of the art methods.

**Questions:**

* Ablation study : I wonder what is exactly the role of the self-attention intra-node mechanism and I am curious of the results if this element is not used ?
* Spatiotemporal positional encoding : there is very few discussion in the article about the importance of the spatiotemporal positional encoding nor analysis of the learned representations, can  you provide some insights on the importance of accurate/not accurate initial spatial information ? It is possible to analyze the learned representations to tell something about the sensors geographic distribution ?

**Strengths And Weaknesses:**

Strengths:
* the paper is well written, easy to follow
* the SOTA is well covered and the proposed approach is well positioned
* I found the proposed model original, clever, and very well explained.
* The novelty is not great as a lot of research has been devoted to the subject of GNN and time series, but the proposed architecture is elegant and intuitive.

Weaknesses :
* Further experiments and analyses would have been useful to assess the real capabilities of the model (see section Questions).

---

> ### Author Response · Authors · 2022-08-02
> **Response to Reviewer PQKX comments**
>
> Thank you for your comments and your positive opinion about our work.
>
> > The novelty is not great as a lot of research has been devoted to the subject of GNN and time series, but the proposed architecture is elegant and intuitive.
>
> Thank you for the good comments about the architecture. Regarding the novelty, we agree that research related to spatiotemporal GNNs is indeed prolific, but we also believe the problem of missing values reconstruction in highly sparse settings to be relevant and nontrivial. We argue that the technical contributions of the paper, supported by experimental results, are an important development for research on this topic.
>
> > Ablation study : I wonder what is exactly the role of the self-attention intra-node mechanism and I am curious of the results if this element is not used ?
>
> Without the self-attention component, each node would be blind to its own time dynamics, thus imputation would be performed only by using information coming from neighbors. The self-attention mechanism can be seen as a way of dealing with a node’s own observed set by using a different set of weights w.r.t. its neighbors (note that we use adjacency matrices without self-loops), since here the objective is to model temporal dependencies only. To show empirically the importance of this component, we added an ablation study in Appendix D to assess the contribution of both the self-attention and cross-attention blocks.
>
> > Spatiotemporal positional encoding : there is very few discussion in the article about the importance of the spatiotemporal positional encoding nor analysis of the learned representations, can you provide some insights on the importance of accurate/not accurate initial spatial information ? It is possible to analyze the learned representations to tell something about the sensors geographic distribution?
>
> In the datasets we consider, the spatiotemporal encoding is always initialized with a learnable embedding. It is difficult to interpret the learned representation as the embedding goes through several nonlinear transformations. However, one of the main purposes of the embeddings is to provide an identification mechanism, which is known to be critical in similar graph-based architectures [1]. Indeed, this allows for learning filters that are parameterized by source and target nodes too.
>
> [1] Dwivedi, Vijay Prakash, and Xavier Bresson. "A generalization of transformer networks to graphs." arXiv preprint arXiv:2012.09699 (2020).

---

> > ### Comment · Reviewer_PQKX · 2022-08-08
> > **Rebuttal feedback**
> >
> > Thank you for the clarifications, my opinion is still positive.

---

### Author Response · Authors · 2022-08-02
**General comments and change list**

We would like to thank again every reviewer for your time and insightful comments. Here you can find a detailed list of the changes we made in the revision of the paper. Thank you for helping us improve the quality of our work.

### Change list

1. Added an experiment on virtual sensing (see Appendix C), i.e., the reconstruction of missing values at sensors with no data during training.
2. Following the suggestion of Reviewers PQKX and tTGX, we added an ablation study on the main components of the spatiotemporal attention layer (see Appendix D).
3. Added an experiment to show the performance of the models with an increasing number of simulated failures, rather than the i.i.d. missing in Table 2 (see Appendix E).
4. As suggested by Reviewers PQtk and K4Xt, we added a discussion on the computational complexities of the most relevant baselines in Appendix A (L600-608).
5. Added reference to relevant works in the context of continuous-time models, following the suggestion of Reviewer tTGX.
6. We added the Appendix in the same file of the paper.
7. We updated the figures according to the suggestions of Reviewer tTGX.
7. Improved readability and clarity
8. Corrected typos

---

### Meta-Review · Area_Chair_bmrY · 2022-08-27

**Recommendation:** Accept
**Confidence:** Certain

**Metareview:**

The paper tackles the important problem of spatiotemporal data imputation via a novel GNN architecture and corresponding spatiotemporal attention mechanism that are both cogent and intuitive.

The author feedback satisfactorily addresses key concerns. In particular the ablation study is a great plus and the virtual sensing results further expand the relevance of the approach.

The AC finds the "auxiliary task learning" setting presented by the authors very interesting and strongly encourages the authors to explore such a setting as future work!

**Award:**

No

---

### Decision · Program_Chairs · 2022-09-14

Accept